# Antitumor Activity and Mechanism of Action of Hormonotoxin, an LHRH Analog Conjugated to Dermaseptin-B2, a Multifunctional Antimicrobial Peptide

**DOI:** 10.3390/ijms222111303

**Published:** 2021-10-20

**Authors:** Mickael Couty, Marie Dusaud, Mickael Miro-Padovani, Liuhui Zhang, Patricia Zadigue, Loussiné Zargarian, Olivier Lequin, Alexandre de la Taille, Jean Delbe, Yamina Hamma-Kourbali, Mohamed Amiche

**Affiliations:** 1INSERM, Institut Mondor de Recherche Biomédicale, Université Paris Est Créteil, F-94010 Créteil, France; mickael.couty@inserm.fr (M.C.); marie.dussaud@gmail.com (M.D.); mickael.miro-padovani@aphp.fr (M.M.-P.); quingfengxu0716@126.com (L.Z.); zadigue@gmail.com (P.Z.); adelataille@hotmail.com (A.d.l.T.); delbe@u-pec.fr (J.D.); hamma@u-pec.fr (Y.H.-K.); 2LBPA, CNRS UMR 8113 École Normale Supérieure Paris-Saclay, 4 Avenue des Sciences, 91190 Gif-sur-Yvette, France; loussine.zargarian@ens-cachan.fr; 3Sorbonne Université, École Normale Supérieure, PSL University, CNRS, Laboratoire des Biomolécules (LBM), 75005 Paris, France; olivier.lequin@upmc.fr

**Keywords:** anticancer drug, antimicrobial peptide (AMP), dermaseptin, frog skin peptides, LHRH, prostate cancer, *Phyllomedusa bicolor*, therapeutic peptides

## Abstract

Prostate cancer is the most common cancer in men. For patients with advanced or metastatic prostate cancer, available treatments can slow down its progression but cannot cure it. The development of innovative drugs resulting from the exploration of biodiversity could open new therapeutic alternatives. Dermaseptin-B2, a natural multifunctional antimicrobial peptide isolated from Amazonian frog skin, has been reported to possess antitumor activity. To improve its pharmacological properties and to decrease its peripheral toxicity and lethality we developed a hormonotoxin molecule composed of dermaseptin-B2 combined with d-Lys^6^-LHRH to target the LHRH receptor. This hormonotoxin has a significant antiproliferative effect on the PC3 tumor cell line, with an IC50 value close to that of dermaseptin-B2. Its antitumor activity has been confirmed in vivo in a xenograft mouse model with PC3 tumors and appears to be better tolerated than dermaseptin-B2. Biophysical experiments showed that the addition of LHRH to dermaseptin-B2 did not alter its secondary structure or biological activity. The combination of different experimental approaches indicated that this hormonotoxin induces cell death by an apoptotic mechanism instead of necrosis, as observed for dermaseptin-B2. These results could explain the lower toxicity observed for this hormonotoxin compared to dermaseptin-B2 and may represent a promising targeting approach for cancer therapy.

## 1. Introduction

Prostate cancer (PCa) is the second most frequent cancer diagnosis made in men and was the fifth leading cause of death worldwide in 2018 [1]. For most men with PCa, their disease will follow an indolent course. The 5-year survival rates are encouraging: 98% and 83% in the USA and Europe, respectively [2]. Localized PCa may be cured with surgery or radionuclide therapy; however, the disease recurs in approximately 20% to 30% of men treated for localized PCa and advanced disease is associated with poor outcomes. Concerning chemotherapy, the current standard treatment for hormone sensitive metastatic PCa is androgen deprivation by luteinizing hormone-releasing hormone (LHRH) agonists or antagonists that induce castration by blocking the gonadotropic axis. Although most men with metastatic PCa initially respond to this androgen-deprivation therapy (ADT), inevitably their cancer progresses on this treatment to a disease state known as castration-resistant prostate cancer (CRPC). Despite the appearance of a panel of new therapies in this indication since 2010, such as second-generation hormone therapy (abiraterone, enzalutamide), taxane-based chemotherapy (docetaxel, cabazitaxel), immunotherapy (sipuleucel-T), targeted therapy (ipilimumab, an anti-CTLA-4 antibody) and metabolic radiotherapy (Ra-223) for men with bone metastasis, the median survival for men with metastatic castration-resistant prostate cancer (mCRPC) is less than 2 years [2]. In this context, the development of innovative therapies therefore represents a major challenge in the hope of offering to patients a more effective treatment, combining low toxicity, and reduced side effects and resistance associated with conventional therapies.

Among these new molecules, those from biodiversity could represent a particular interest. A group of interesting peptides from natural sources are antimicrobial peptides (AMPs) [3]. Indeed, in recent years, an increasing number of articles show that these AMPs are in fact multifunctional peptides such as anticancer agents, immunomodulators, chemokines, vaccine adjuvants, or regulators of innate defense [4,5,6]. From this group of AMPs, the cationic antimicrobial peptides (CAPs) offer a concrete development path derived from biodiversity. Characterized for nearly thirty years, they were initially studied for their antimicrobial virtues. Indeed, a growing number of studies have shown that some of these peptides, which are toxic to bacteria but not to normal mammalian cells, have a broad spectrum of cytotoxic activity against cancer cells. Electrostatic interactions between the positive charges carried by CAPs and the anionic components of cell membranes are considered to be the major elements involved in the selective destruction of cancer cells [7]. Among CAPs, dermaseptins B2 and B3 (DRS-B2 and DRS-B3) are two natural antimicrobial peptides isolated from the skin of an Amazonian tree frog of the genus *Phyllomedusa bicolor* [8,9]. Our team initially reported significant antitumor activity of DRS-B2 and DRS-B3 on various human cell lines including prostate cancer, while they had no effect on normal cells [5,10]. Furthermore, the antitumor activity of DRS-B2 was confirmed in vivo in a PC3 prostatic tumor cell line xenografted in an athymic mouse model. Treatment with DRS-B2 at 2.5 mg/kg body weight six times a week in peritumoral mice reduced the tumor growth 50% after 35 days. Concerning the mechanism of action of DRS-B2, our previous studies suggested a rapid mechanism of cell death, with aggregation at the plasma membrane of cancer cells and penetration into the cytoplasm and the nucleus [5,11].

These polycationic peptides, that appear to interact and specifically cross the membrane of the tumor cells, with no effect on normal cells, represent an innovative technological platform for the development and design of original molecules that can be used in the targeted treatment of cancers resistant to current therapies. However, its use in vivo could raise the problem of significant toxicity regarding the doses used. To reduce the toxicity of the pharmacological molecule, a targeted therapy could represent a promising way. The concept of a hormonotoxin (H-B2) was born from this problem, based on tumor targeting by associating DRS-B2 with a hormone (H), the receptor if which is overexpressed on the tumor surface. This concept is related to the immunotoxin approach, which combines a toxin with a monoclonal antibody [12]. The specific interaction of the ligand with its receptor would ideally allow targeting of tumor cells, while optimizing the interaction of the peptide with the membrane. The advantage of peptides is the simplicity of their production by chemical synthesis. Numerous studies have shown that many membrane receptors are overexpressed on the surface of cancer cells and that there are natural or synthetic peptide ligands (agonists or antagonists) that bind to them with very good affinity and selectivity [13]. This is the case for the luteinizing hormone-releasing hormone receptor (LHRH-R), the expression rate of which is greater than 80% in endometrial, ovarian and prostate cancer cells [14]. Available data strongly suggest that about 91% of prostate cancers express LHRH-R high-affinity binding sites [15]. In these cancers, in vitro proliferation may be inhibited by analogs of LHRH in a dose- and time-dependent manner [15,16,17]. As a result, LHRH-R appears to be an ideal target for the development of personalized therapeutic treatment of various human cancers. An example is the fusion of LHRH or its analogs with various bacterial and plant toxins that have been used to target and kill cancer cells expressing LHRH receptors [18,19,20,21,22].

In this paper we report the design and synthesis of the chimeric peptide H-B2, a molecule composed of dermaseptin-B2 coupled to an LHRH analog, and studies of its structure and biological activities in vitro and in vivo, as well as deciphering its antitumor mechanism of action.

## 2. Results

The hormonotoxin H-B2 and DRS-B2 induce a significant antiproliferative effect on the hormone-resistant prostate tumor cell line PC3 and have low hemolytic activity.

In this study we used four different prostate cell lines (PC3, DU145, 22RV1 and BPH-1) which are presented in Table 1.

The PC3, DU145, and 22Rv1 cell lines were used as models for androgen-independent prostate cancer. PC3 and DU145 were derived from bone and brain metastasis, respectively. The 22Rv1 cell line was originally derived from the primary site of an advanced PCa that was serially transplanted in nude mice, and BPH-1 is a prostatic hyperplasic cell line. The primary structure of the synthetic hormonotoxin H-B2 is shown in Figure 1 and its RP-HPLC profile and ESI-MS spectra are reported in the Appendix A.

The effect of synthetic H-B2 and DRS-B2 after 48 h of treatment was evaluated on different prostate cell lines. Results show that the viability of all cell lines was reduced dose-dependently by DRS-B2 and H-B2 (Figure 2).

The concentrations of peptides required to inhibit cell proliferation by 50% (IC_50_) are reported in Table 2.

Since PC3 cells present a relatively good sensitivity to H-B2 and DRS-B2, we decided to use this cell line for further investigations concerning the in vivo study of H-B2 and exploration of its antiproliferative mechanism of action.

Since CAPs could present hemolytic activity, H-B2 and DRS-B2 were tested on human red blood cells. The hemolytic activity was determined by incubation of DRS-B2 or H-B2 at different doses with human erythrocytes. In parallel, the erythrocytes were incubated in the presence of 0.2% (*v*/*v*) Triton representing the positive control (100% hemolysis). Results presented in Figure 3 show that DRS-B2 and H-B2 have low hemolytic activity.

Indeed, while some cationic antimicrobial peptides show hemolytic activity, DRS-B2 and H-B2 do not cause more than 10% hemolysis at the highest doses tested (50 µM) and could be considered as low cytotoxicity peptides. This finding is encouraging and essential for in vivo testing and for potential therapeutic use.

### 2.1. Hormonotoxin H-B2 Significantly Inhibits Tumor Growth in Xenografted Mice without Measurable Side Effects

To investigate the antitumor efficacy of H-B2, we performed in vivo experiments using xenografted PC3 tumor cells in nude mice. Fourteen days after injection of PC3 cells, tumors of around 100 mm^3^ developed. The mice were then randomized into four groups (*n* = 6 mice/group) and treated by IP injection of 2.5 and 5 mg/kg of H-B2 or 2.5 mg/kg of DRS-B2 or vehicle, three times per week for five weeks. The results showed that H-B2 inhibited PC3 tumor growth in a dose-dependent manner (Figure 4). The inhibition of the tumor growth was about 35% and 54% when mice were treated with H-B2 at 2.5 mg/kg and 5 mg/kg, respectively (Figure 4A). At the dose of 2.5 mg/kg, H-B2 had a better antitumor effect (35%) than DRS-B2 (26%) (Figure 4A). Similar results were obtained by analyzing the weights of tumors harvested from tumor-bearing mice (Figure 4B).

To further define the effect of H-B2 on tumor proliferation in vivo, Ki67 labeling was performed on tumor sections from each group. As shown in Figure 5, the proliferation index was significantly inhibited in H-B2-treated mice (35% inhibition at 2.5 mg/kg and 68% at 5 mg/kg) as compared to the control group.

Taken together, these results indicate that H-B2 has a better antitumor effect than DRS-B2 and less toxicity in mice (data not shown).

### 2.2. Mechanism of Antitumor Action of the Hormonotoxin H-B2 in Comparison with DRS-B2

The mechanism of antitumor action of these two peptides was addressed by studying, on the one hand, their bioactive structure by three spectroscopic approaches: CD, fluorescence, and 2D-NMR in the presence of micelles mimicking the plasma membranes of the target cells; and on the other hand, by measuring: the cell viability after double annexin V-FITC and PI staining, the cytotoxic activity by measuring the cytoplasmic LDH released, and the DNA fragmentation induced by these peptides.

Addition of the hormonal analog (d-Lys^6^-LHRH) to DRS-B2 did not alter the secondary structure of the hormonotoxin H-B2.

Preliminary indications of the secondary structures of these peptides, obtained by CD measurements in PBS and in zwitterionic detergent (DPC) at different concentrations, are shown in Figure 6.

The CD spectrum of DRS-B2 and H-B2 in PBS buffer showed that these peptides have very little ordered structure. However, there is a clear change in the spectra of these two peptides as soon as the critical micellar concentration (CMC) of DPC is reached (Figure 6(A1) for DRS-B2 and Figure 6(B1) for H-B2) that indicates an enhanced helical content, with minima at 208 and 222 nm. Deconvolution of the spectra allows us to quantify the relative proportions of the secondary structures of our peptides. Thus, in the absence of micelles in the medium, the observed α-helix ratio is about 11% for DRS-B2 and 10% for HB2 as shown in Figure 6(A2) and Figure 6(B1), respectively. When the concentration of DPC is increased to 5 mM, the α-helicity rates then reach 25% and 55% for H-B2 and DRS-B2, respectively.

The presence of a W residue in the sequence of DRS-B2 and H-B2 allows us to study the influence of the microenvironment on the structure of these two peptides. Indeed, the fluorescence emission maximum of W is between 320 and 355 nm when excited at 290 nm, and the maximum emission wavelength reflects the exposure of W to the solvent. This fluorescence is measured in an aqueous solution (PBS 1x) for observation in a nonstructural environment (the peptide does not form an α-helix in water) and in a micellar solution to study the effect of a lipid-like microenvironment (Figure 6(A3,B3)). We observe that beyond 1 mM, which is the CMC of DPC, the fluorescence emission maxima of DRS-B2 and H-B2 shift to shorter wavelengths (“blue shift”) and show a strong increase in fluorescence intensity (hyperchromic shift). These spectral changes reflect a change from a hydrophilic to hydrophobic environment that can be explained either by the burial of the W residue within the hydrophobic layers of DPC micelles, or by burying the W after folding of the peptide following conformational changes.

Finally, the structure of H-B2 was investigated using NMR in a micellar DPC solution. This technique allows a direct analysis of peptide conformation and flexibility at the residue level. For each residue of the peptide chain, the chemical shift deviation (CSD) of the Hα protons were calculated (Figure 7A).

This corresponds to the difference between the observed chemical shift of the Hα protons and the “random coil” chemical shift (obtained in the absence of structuring and corresponding to values taken from a library of short unstructured peptides). CSD values which deviate from 0 ppm (<−0.1 or >+0.1 ppm) are typically used to detect the presence of secondary structures. In the DRS-B2 portion of hormonotoxin H-B2, most residues exhibit significantly negative Hα CSD values (average −0.23 ppm), indicating the presence of stable helical conformations, as evidenced by CD. H-B2 forms an α-helix along virtually the entire length of the DRS-B2 part (residues 2–31), with greater stability in the first half of the helix. The CSD values are very close to those observed for DRS-B2 alone, proving that the C-terminal attachment of the LHRH segment does not affect the helical structure of the DRS-B2 segment. On the other hand, the LHRH segment appears to be less structured, with possibly the formation of turn structures.

The linewidth of HN protons was also analyzed by measuring the intensity of ^15^N-^1^H cross peaks on the natural abundance ^15^N-^1^H 2D HSQC spectrum (Figure 7B). The proton linewidth is influenced by peptide dynamics and interaction with the larger molecular weight DPC micelle. The LHRH segment shows a different behavior from the DRS-B2 segment in H-B2, with stronger ^15^N-^1^H cross peak intensities. This shows that the LHRH segment is more dynamic and does not appear to strongly interact with the micelle.

### 2.3. The Cell Death Induced by H-B2 and DRS-B2 Uses Two Different Mechanisms That Are Related to Apoptosis and Necrosis, Respectively

To obtain more information about how H-B2 acts on PC3 tumor cells, the antiproliferative mechanism of action of this peptide was studied in comparison with that of DRS-B2. Since the known antiproliferative effect of DRS-B2 on tumor cells resemble more of a membrane killing-like effect, experiments concerning the cytotoxic effect, and the induction of apoptosis or necrosis were investigated with H-B2 in comparison with DRS-B2.

A key feature of cells undergoing apoptosis, necrosis, and other forms of cellular damage could be analyzed by measuring the activity of cytoplasmic lactate dehydrogenase (LDH) released by damaged cells. Previously, we have shown that DRS-2 increased this release [11]. Thus, PC3 cells were treated with different concentrations (1, 5, 7.5 and 10 µM) of H-B2 or DRS-B2 for 24 h and the amount of released cytoplasmic LDH into the medium was measured (Figure 8).

Cells treated with Triton X100 0.9% (*v*/*v*) were used as an internal positive control. The results show a low release of cytoplasmic LDH when the PC3 cells are treated with 1 µM of H-B2 or DRS-B2, with 15% and 10% release, respectively. The LDH release is maximal upon addition of 5 µM or higher doses of these peptides. However, this maximum release was limited to 65–70%. These data suggested that the cytotoxic effect of H-B2 on PC3 cells is comparable to that of DRS-B2.

To further characterize the effect of H-B2 and DRS-B2 on cell death in PC3 cells, we first examined the viability of cells treated with these peptides by flow cytometry using annexin V-FITC and PI double staining of the cells. As shown in Figure 9, flow cytometry analysis showed that H-B2 promoted cell apoptosis in a dose-dependent manner.

Histogram of each population obtained by flow cytometry in the different conditions are presented in the Appendix A and the percentages of cells in each population are reported in the Table 3.

After being treated with H-B2 for 24 h, 4%, 15%, and 63% of double staining A-V^+^/PI^−^ which corresponds to early apoptotic cells were found in PC3 cells treated with 1 μM, 2.5 μM, and 5 μM, respectively. These values were significantly higher than those of untreated cells (1%) and similar (63.5%) to cells treated with H_2_O_2_ used as a positive control. DRS-B2 presented a lower effect with only 5.5%, 4% and 9.5% of double staining A-V+/PI- when PC3 cells were treated with DRS-B2 at 1 μM, 2.5 μM and 5 μM, which agrees with results described by Van Zoggel et al. [11].

DNA fragmentation represents the final stage of apoptosis. To examine apoptosis through DNA fragmentation induced by H-B2 or DRS-B2, we used the terminal deoxynucleotidyl transferase (Tdt) dUTP Nick-End Labeling (TUNEL) assay, in which DNA strand breaks are detected by enzymatic labeling of free 3′-OH ends with modified nucleotides. The analysis of DNA fragmentation in PC3 cells by the TUNEL technique shows that the percentage of fragmented DNA increases with H-B2 treatments. Treatment of PC3 cells with 5 µM H-B2 induces a DNA fragmentation level of 69% compared to untreated cells (1.5%), whereas it reaches only 9% when treated with the same concentration of DRS-B2. (Figure 10).

PC3 cells treated with 10 nM Taxotere or 1 µM of Staurosporine as positive controls show 96% and 86% DNA fragmentation, respectively (Figure 10). These data confirm that the mechanism of PC3 cell death induced by H-B2 is different from that induced by DRS-B2, which is necrotic, and could rather correspond to an apoptotic mechanism, in agreement with the flow cytometry results described previously.

## 3. Discussion

With 1.1 million prostate cancers diagnosed worldwide in 2012, of which approximately 6% are metastatic from the start, the development of treatments for mCPRC is a global necessity with a promising market. The development of treatments for the management of castration-resistant metastatic patients is necessary because currently available treatments only allow a few months of survival gain, mainly due to problems of drug resistance of cancer cells. In this context, the use of natural or synthetic peptides could offer interesting therapeutic approaches. To limit peripheral toxicity and increase local concentrations, numerous cytotoxic molecules conjugated with peptide hormones such as LHRH or somatostatin, the receptors of which are widely overexpressed on the surface of tumor membranes, have been developed [23,24,25,26,27]. Currently, only one cytotoxic peptide is in clinical development (Phase 2 completed), AEZS-108 (zoptarelin doxorubicin) [28].

Starting from our previous studies on the anticancer activity of the CAP dermaseptin-B2 (DRS-B2), a natural peptide issued from biodiversity, we have developed a synthetic chimeric H-B2 peptide combining DRS-B2 and LHRH peptides [5,10,11]. The use of LHRH was justified by the fact that 86% of prostate cancers express the LHRH receptor [14]. A peptide targeting this receptor therefore should increase the specificity for cancer cells and decrease the in vivo toxicity of DRS-B2.

At the structural level, the coupling of the peptide hormone (H) LHRH to the C-terminus of DRS-B2 does not modify the structure of DRS-B2 nor its conformational behavior in a membrane environment. The characteristics essential to the cytotoxic mode of action of these polycationic peptides and to the targeted molecular interaction are therefore preserved. CAPs are supposed to act on cancer cells in which the outer layer of the plasma membrane is highly negatively charged, as for bacterial plasma membranes. In the case of cancer cells, the negatively charged membrane is due to the presence of phosphatidylserine, negatively charged mucin proteins or highly sulfated GAGs (5, 25). After binding, dermaseptin peptides accumulate in a carpet-like manner on the outside of a lipid bilayer until a threshold concentration is reached, causing them to form pores in which the peptides are inserted with the phospholipid headgroups of the membrane [29,30,31,32,33,34]. We have shown that H-B2 was structured as a continuous helix in its “toxin” portion (DRS-B2) and that the “hormone” ligand portion remains free to interact with LHRH-R.

In vitro, H-B2 was slightly more effective than DRS-B2 on the proliferation of the various prostate cancer cell lines tested. We could also mention that cells expressing high levels of LHRH receptors, such as PC3 and DU145, presented a modest but significant gain in IC_50_ when compared with those of DRS-B2 alone. Targeting cancer cells with an overexpressed receptor such as LHRH is a strategy that could permit improvement to the efficiency of a peptide such as DRS-B2 to gain in its ED_50_. However, in the case of H-B2, this gain is low. We could imagine that even if the LHRH receptor is overexpressed on PC3 cells, its concentration at the surface of the plasma membrane is too low to guide enough DRS-B2 to the membrane to complete its carpet-like structure and further form pores in the plasma membrane. As saturation of the LHRH receptor by H-B2 may not be sufficient to kill cells, an additional part of DRS-B2 in the H-B2 structure may be necessary to do it.

In vivo, H-B2 inhibited tumor growth by more than 50% and significantly decreased proliferation without any major side effects. Since the addition of the hormone peptide to DRS-B2 permits a better tolerance when injected in mice, we could conclude that despite significant improvement in the IC_50_ on the cell line, the hormone peptide improved the in vivo activity of DRS-B2, as we could treat mice with 5 mg/kg H-B2 without side effects instead of 2.5 mg/kg with DRS-B2. It is interesting to note that the treatment of prostate tumor cells with H-B2 is globally well tolerated. We did not observe any major abnormalities in the blood workup after H-B2 injection. Importantly, at a dose of 10 mg/kg, DRS-B2 is lethal while H-B2 at the same dose was well tolerated in toxicity tests, with no behavioral changes in the mice or weight loss. The design of the H-B2 chimeric peptide, thus made it possible to circumvent the toxicity of DRS-B2 while maintaining its antitumor efficacy.

Concerning the mechanism of action, the combination of different experimental approaches used in this study, such as cell viability by flow cytometry, cytotoxicity by cytoplasmic LDH release, and DNA fragmentation by TUNEL assay, allowed us to show that the mechanism of cell death induced by H-B2 is probably different from that of DRS-B2 and might be similar to apoptosis. Indeed, we showed that both peptides have the same effect on cytoplasmic LDH release but a distinctly different cell labeling when cell viability is analyzed by flow cytometry and DNA fragmentation by TUNEL assays. The differences between the effect of the two peptides observed in the TUNEL assay show that H-B2 induces DNA fragmentation up to 70% in contrast to DRS-B2 which is less than 10%. These results are also in agreement with those obtained in 2012 by Van Zoggel et al. who showed a double Annexin V+/PI+ labeling of PC3 cells treated with DRS-B2, suggesting a necrotic cell death mechanism [11].

Apart from DRS-B2 previously studied in the laboratory, two new members of the dermaseptin family have been recently reported to exhibit antitumor activities: Dermaseptin-PP from *Phyllomedusa palliata* and Dermaseptin-PT9 from *Phyllomedusa tarsius* [35,36]. Both exhibited antiproliferative activity against various human tumor cells, rapid LDH release activity, and an apoptosis-like cell death mechanism. Among these dermaseptins, DRS-B2 presented the higher ED_50_ on various tumor cells.

Finally, the study of the hemolytic effect of the different peptides allowed us to observe that both DRS-B2 and H-B2 have low hemolytic activity, which is encouraging in the perspective of a therapeutic approach. Further in vivo studies on mice could be necessary to obtain more pharmacokinetic and toxicology information to complete the study.

## 4. Materials and Methods

Peptide synthesis. The LHRH analog containing a dK residue (d for D configuration of the Lys residue) in position 6 (peptide A: pEHWSY(dK)LRG-amide) and DRS-B2 (peptide B: GLWSKIKEVGKEAAKAAAKAAGKAALGACSEAV-acid) were synthesized by X′PROCHEM Company (Lille, France). The chimeric peptide H-B2 (Figure 1A), composed of the peptide A grafted by its epsilon NH_2_ of the dK^6^ residue with the COOH-terminal of the peptide B, is determined to be >95% pure by RP-HPLC (Appendix A) and its molecular weight determined by ESI-mass spectrometry is [M + 4H]/4 = 1105.2 (Appendix A). The peptide was dissolved in sterile water for in vitro biological studies and in adequate buffer for the in vivo and structural analysis.

Circular dichroism (CD) spectroscopy of peptides. The helical structures of HB2 and DRS-B2 were analyzed by CD spectroscopy using a Jobin Yvon CD6 dichrograph linked to a PC microprocessor as described in [34]. Briefly, measurements were calibrated with (+)-10-camphorsulfonic acid and performed with 10 μM H-B2 or DRS-B2 diluted in PBS alone or with increasing concentrations of dodecylphosphocholin (DPC) (10, 30, 100, 1000 and 5000 μM) at 25 °C using a quartz cuvette (Hellma) with a path length of 0.1 cm. Spectra, recorded in 1 nm steps, were averaged over five scans, and corrected for the baseline. The CD spectra were deconvoluted using CDNN Software [37]. Circular dichroism measurements are reported as Δε/*n*, where Δε is the dichroic increment (M^−1^ cm^−1^) and *n* is the number of residues in the peptide. The α-helix content of peptides was obtained using the relation: Pα = −[Δε_222nm_ × 10] (Pα: percentage of α-helix; Δε_222nm_: dichroic increment per residue at 222 nm) [38].

Fluorescence of tryptophan-containing peptides was performed as previously described by Dos Santos et al. [5]. Emission spectra were recorded on a Jobin-Yvon Fluoromax II instrument (HORIBA Jobin-Yvon, Montpellier, France) equipped with an Ozone-free 150 W xenon lamp. The excitation wavelength was 290 nm, and the emission spectra were acquired at 300–360 nm. At least five measurements for each titration point were recorded with an integration time of 1 s. The H-B2 and DRS-B2 concentration in PBS was 2 μM, and the DPC concentration varied from 0 to 1000 μM. Tryptophan fluorescence was determined by subtracting spectra without the peptide.

Nuclear Magnetic Resonance (NMR) Spectroscopy. The NMR sample was prepared in 550 μL of H_2_O/D_2_O (90:10 *v*/*v*) in the presence of 85 mM DPC-d_38_ (d_38_-dodecylphosphocholin, MAPCHO^®^-12-d_38_, Avanti Polar Lipids) and using a peptide concentration of 1 mM. The pH was set to 4.1 using microliter amounts of NaOH 0.1 M. DSS was added at a concentration of 0.11 mM for chemical shift calibration. The NMR spectra were acquired at 45 °C on a 500 MHz (11.7 T) Avance III Bruker^®^ spectrometer equipped with a TCI cryoprobe. The following experiments were recorded: 2D ^1^H-^1^H TOCSY (66 ms mixing time), 2D ^1^H-^1^H NOESY (150 ms mixing time), 2D natural abundance ^15^N-^1^H and 2D ^13^C-^1^H HSQC experiments, as described in [34].

Cell culture. Human PCa cell lines, PC3, DU145 and 22Rv1 were purchased from ATCC (American Type Culture Collection) and were cultured in RPMI supplemented with 10% of fetal bovine serum (FBS). The human hyperplasic cell line BPH1 was purchased from the German Collection of Cell Cultures (DSMZ, Braunschweig, Germany). BPH1 were maintained in RPMI 1640 supplemented by 10% FBS, 20 ng/mL testosterone and 1% insulin-transferrin-selenium. Cell cultures were maintained at 37 °C and 5% CO_2_ in a humidified atmosphere. All culture reagents were purchased from Life Technologies (Cergy-Pontoise, France). All experiments were performed as previously described by Dos Santos et al. [5]

Cell viability assays. Cells were seeded at a density of 5 × 10^3^ cells/well in 96-multiwell plates in complete medium and incubated for 24 h at 37 °C in a controlled humidified 7% CO_2_ environment. Cells were then treated with DRS-B2 or H-B2 as indicated, for 48 h. Cell viability was measured using the 3-(4,5-dimethylthiazol2-yl)-diphenyltetrazolium bromide (MTT) dye method (Sigma, Saint Quentin Fallavier, France) according to the manufacturer’s instructions. Each experiment was performed in triplicate with at least three independent experiments, as previously described by Dos santos et al. [5]. IC_50_ values were determined by GraphPad Prism 5.0 (GraphPad Software San Diego, CA, USA).

Hemolysis assessment. The hemolytic activity of H-B2 and DRS-B2 was determined using fresh human erythrocytes from a healthy donor that was prepared as follows: 4 mL of whole blood was collected and centrifuged at 900× *g* for 10 min at 4 °C to separate plasma from erythrocytes. The pellet was then rinsed with PBS pH 7.4 and centrifuged at 900× *g* for 10 min at 4 °C. After counting and making a red cell solution at 4 × 10^8^ erythrocytes/mL (diluted in PBS pH 7.4), 50 µL of a diluted peptide solution was added in a cascade to which 50 µL of the erythrocyte solution was added (made in triplicate). After 1 h of incubation at 37 °C, the tubes were centrifuged at 12,000 rpm for 15 s at 4 °C. The supernatant was then recovered. The hemoglobin present in the erythrocytes was determined in the supernatants via a plate reader at 450 nm. A parallel incubation in the presence of 0.2% (*v*/*v*) Triton was carried out to determine the absorbance associated with 100% hemolysis.

Tumor xenograft studies were performed as previously described by Van Zoggel et al. [11]. PC3 (2 × 10^6^) cells were injected subcutaneously into the right flank of 4-week-old male NMRI nude mice (Janvier, Le Genest-Saint-Isle, France). When the tumor volume reached approximately 100 mm^3^, the mice were randomly divided into four groups (*n* = 6): control (PBS), DRS-B2 (2.5 mg/kg), H-B2 (2.5 mg/kg), H-B2 (5 mg/kg) by intraperitoneal injection twice a week. Tumor size was measured two times per week with a caliper and the tumor volume was calculated with the formula: V = 4/3π × R1^2^ × R2 with radius 1 (R1), and radius 2 (R2). At the end of the experiment, the mice were sacrificed, and their body weight was measured. The tumors were isolated, weighted and then fixed in formalin.

Immunohistochemical analysis of the tumors was performed as previously described by Van Zoggel et al. [11]. The fixed tumors were embedded in paraffin and then 6 µm sections were prepared. The tumors sections were deparaffinized, antigen unmasking was performed, and endogenous peroxidase activity was inactivated with a 2% hydrogen peroxide solution for 10 min. Unspecific staining was blocked using Power Block Universal reagent (Biogenex Laboratories/Microm Microtech, Francheville, France) for 10 min at 37 °C. Tissues were then incubated 2 h at room temperature with anti-human Ki67 antibody (Mouse monoclonal, M7240, Dako, 1:50) for 2 h. Immuno-complexes were revealed using HRP conjugated secondary antibodies and the DAB substrate. Tissues were then counterstained with hematoxylin and cover slipped with Mowiol mounting medium. Quantification of Ki67 positive stained cells was quantified by image J software analysis on the whole tumor section.

Lactate dehydrogenase (LDH) release assay. The cytoplasm LDH release assay was performed as previously described in [11]. Briefly, PC3 cells were grown in a 96 well plate (1500 cells/well/100 μL) in complete medium and treated with various concentrations of H-B2 or DRS-B2. Cell membrane integrity was evaluated by measuring the LDH activity released into the culture media 24 h after peptide exposure. The CytoTox96 nonradioactive cytotoxicity assay (Promega; Charbonnières-les-Bains, France) was performed according to the manufacturer’s instructions and quantified by measuring the absorbance at 490 nm. The 100% cytotoxicity corresponded to the LDH released with treatment of the cells with Triton X100 at 0.9% (*v*/*v*).

Apoptosis analysis by flow cytometry. For the apoptosis assay, an FITC-Annexin-V (A-V) and Propidium Iodide (PI) double staining method was used. PC3 cells were grown in a 12-well plate and treated or not with H-B2, DRS-B2 or hydrogen peroxide as described previously [5]. At 24 h after exposure, cell viability was evaluated by flow cytometry analysis with FITC-Annexin-V (A-V) and propidium iodide staining. Medium and trypsinized cells were collected and washed with PBS. After centrifugation, cells were suspended in PBS to obtain a cell density of 0.5 × 10^6^ cells per mL. One milliliter of this cell extract was centrifuged, suspended in 200 μL PBS, transferred to a microtiterplate with a round bottom and centrifuged again. The resulting cell pellet was resuspended in 200 μL of Binding Buffer 1x (BD PharmingenTM), containing 5 μL of FITC-Annexin-V (BD PharmingenTM) and incubated for 10 min in the dark at room temperature. The cells were washed with PBS and incubated with 200 μL of Binding Buffer 1x containing PI (final concentration 1 μg/mL) (BD Pharmin-genTM)) for 5 min in the dark at room temperature. Flow cytometry analysis was performed with an LSR Fortessa X20 analyzer (BD Biosciences, Franklin Lakes, NJ, USA) and FlowJo V10 software.

Terminal transferase-mediated dUTP Nick End-Labeling (TUNEL) assay. PC3 cells were cultured in 6-well plates at 150,000 cells per well in complete medium and treated or not with 1 or 2.5 µM of DRS-B2 or H-B2 and with LHRH (5 μM) or Taxotere (10 nM) and Staurosporine (1 μM) as a positive control. After 24 h of treatment, the supernatant was aspirated and the cells trypsinized and plated on a Superfrost PLUS slide and fixed with a 4% formaldehyde solution without methanol. The cells were then permeabilized with a 0.2% Triton X-100 solution in 1x PBS for 5 min at room temperature. Terminal transferase-mediated dUTP nick end-labeling (TUNEL) was performed according to the DeadEnd™ Fluorometric TUNEL System kit from Promega^®^. In addition to the fluorescein labeling, a second labeling with DAPI, intercalating DNA was performed by incubating for 15 min at room temperature with a 1 µg/mL solution. The different labeling steps were performed avoiding any exposure to light.

Imaging was performed with an Axiolmager M2 epifluorescence microscope. The images obtained by microscopy were recorded with Zen 2012 software and analyzed with ImageJ© software. Each image underwent a thresholding of fluorescence calibrated on the positive control, and which was preserved for all the other images. The software then analyzed the number of DAPI and fluorescein fluorescence events to calculate the ratio between the total number of DAPI-labeled cell nuclei and the number of fluorescein-labeled nuclei with fragmented DNA. The experiment was performed in triplicate.

Statistical analysis. The statistical analyses were performed using GraphPad PrismTM version 4.00 software from GraphPad Software Inc. (San Diego, CA, USA). The results are expressed as the means ± standard deviation (SD) or standard error of the mean (SEM) of at least three determinations for each test from three independent experiments. Statistical analyses were carried out using the unpaired *t*-test. The statistical significance of the differences is given as * *p* < 0.05; ** *p* < 0.01; *** *p* < 0.001; ns: not significant.

## Figures and Tables

**Figure 1 ijms-22-11303-f001:**
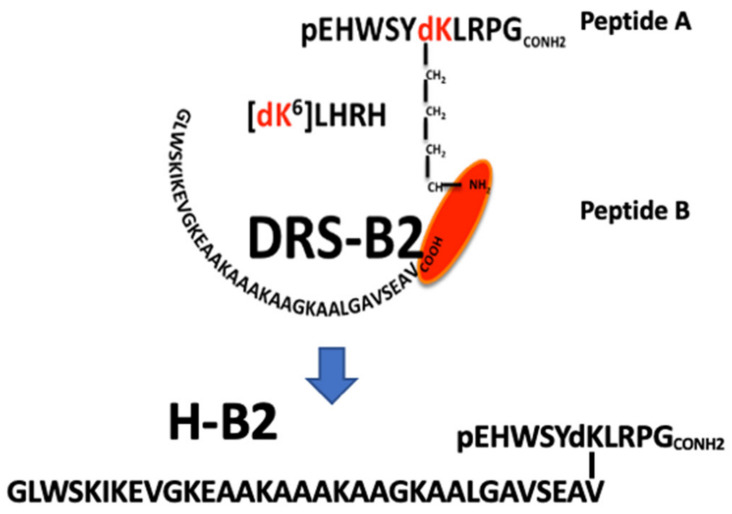
Synthesis of the chimeric H-B2 peptide. Primary structure of the hormonotoxin [D-Lys^6^]-LHRH-dermaseptin B2 (H-B2). The amino acid sequence of dermaseptin B2 is represented as peptide B and the amino sequence of the LHRH analog with a D-Lys (dK) in position 6 is represented as peptide A.

**Figure 2 ijms-22-11303-f002:**
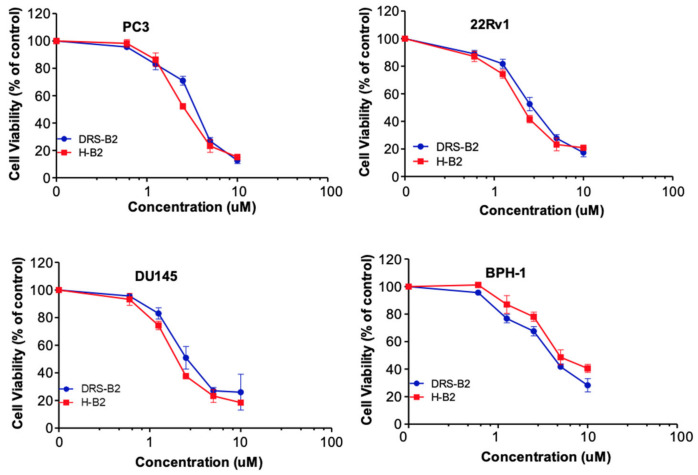
Sensitivity of human PCa cell lines to H-B2 is dependent on LHRH-R expression. The dose–response effect of H-B2 and DRS-B2 (0.1 to 100 μM) on the cell viability of PC3, DU145, 22Rv1 and BPH-1 cells was evaluated using MTT assay. The 100% of cell viability corresponds to cells without treatment.

**Figure 3 ijms-22-11303-f003:**
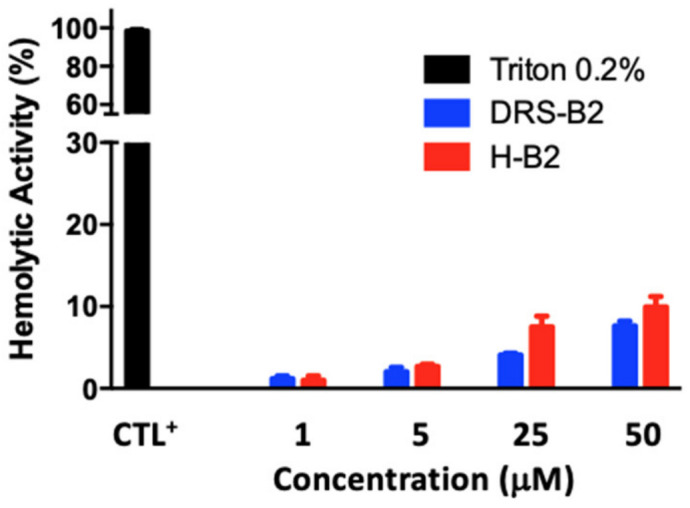
Hemolytic activity of DRS-B2 and H-B2 on human erythrocytes. Human erythrocytes were cultured with different concentrations of DRS-B2 or H-B2 for 1 h at 37 °C. Cells treated with 0.2% Triton X100 were used as a positive control (CTL^+^) and correspond to 100% of the hemolytic activity used as reference.

**Figure 4 ijms-22-11303-f004:**
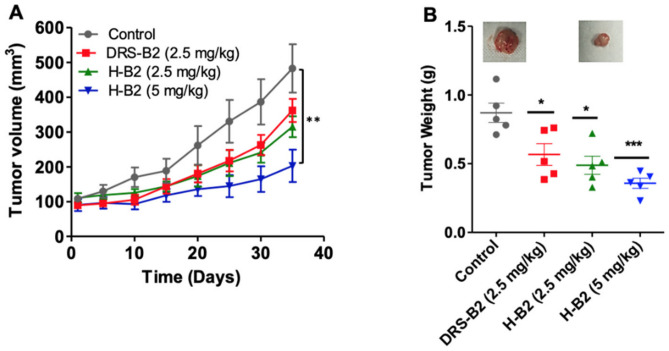
Effect of DRS-B2 and H-B2 on tumor growth in vivo. Nude mice were subcutaneously injected with PC3 cells and treatments began after the xenograft tumor reached 100 mm^3^. Mice were treated twice a week intraperitoneally with PBS (Control), DRS-B2 (2.5 mg/kg) or H-B2 at 2.5 mg/kg or 5 mg/kg. (**A**) Effect of tumor volume versus time of treatment. (**B**) Tumor weight after sacrifice of mice. Data are presented as mean ± SEM. Differences were considered significant at *p* < 0.05 (*), *p* < 0.01 (**) and *p* < 0.001 (***).

**Figure 5 ijms-22-11303-f005:**
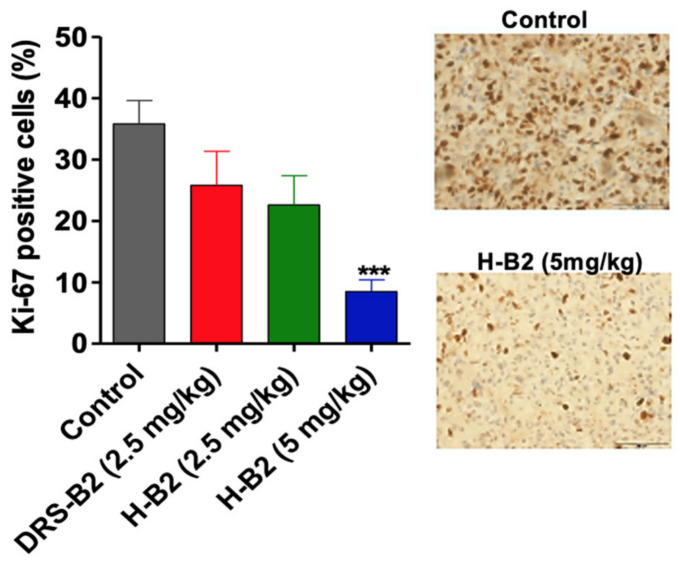
Effect of DRS-B2 and H-B2 on tumor proliferation in vivo. PC3 tumor proliferation was evaluated by Ki67 staining of frozen tissue sections. Proliferation was quantified by image J software analysis of Ki67 positive stained cells on the whole tumor section. The data are mean areas ± SEM, *p* < 0.001 (***). Representative tumor section, scale bar, 50 µm.

**Figure 6 ijms-22-11303-f006:**
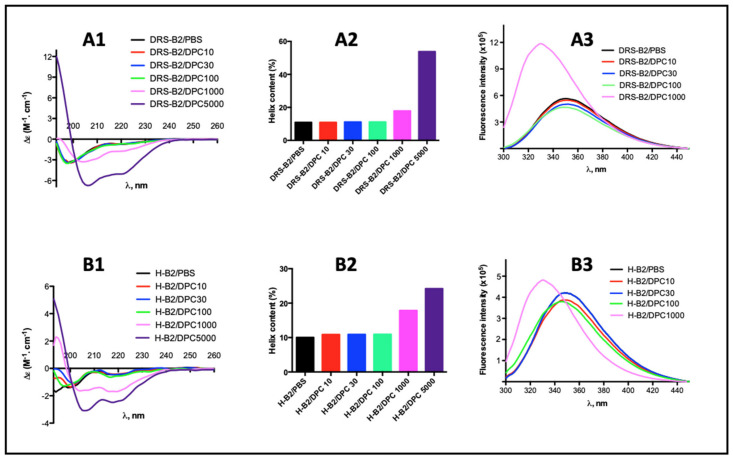
Circular dichroism and fluorescence spectra of DRS-B2 (**A1**,**A3**) and H-B2 (**B1**,**B3**) alone in PBS or in the presence of increasing concentration of DPC. (**A2**,**B2**) represent the percent of helix content in DRS-B2 and H-B2, alone or in presence of an increasing concentration of DPC, respectively.

**Figure 7 ijms-22-11303-f007:**
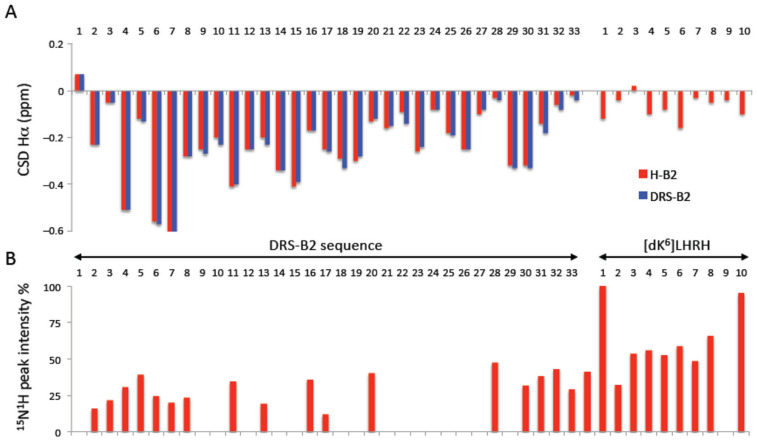
Structure of H-B2 and DRS-B2 analyzed by NMR. (**A**) Chemical shift deviations (CSDs) of Hα protons of DRS-B2 and H-B2 in the presence of DPC. The CSD values were calculated as the difference between observed chemical shifts and random coil chemical shifts. (**B**) Intensity of ^15^N-^1^H peaks on 2D ^15^N-^1^H HSQC. The intensities were normalized to the highest intensity peak. Only well resolved, unambiguously assigned peaks were considered.

**Figure 8 ijms-22-11303-f008:**
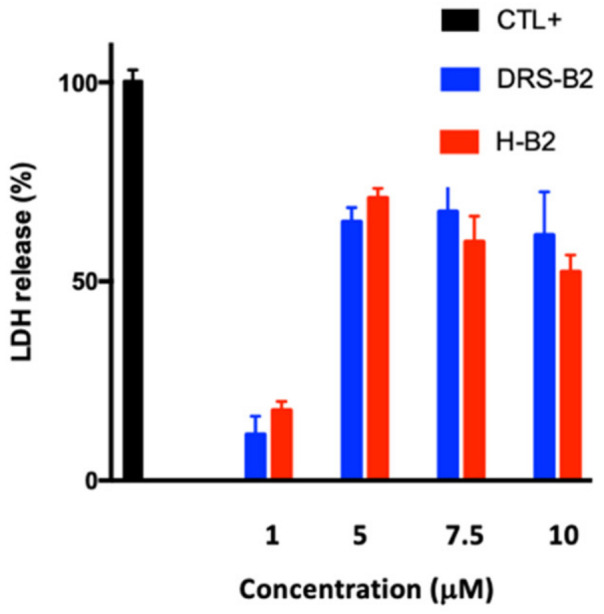
Cytotoxic effect of DRS-B2 on PC3. Twenty-four hours after plating, PC3 cells were single treated with different concentrations of DRS-B2 or H-B2. Triton X100 at 0.9% (*v*/*v*) was used as an internal positive control. LDH release was measured with a CytoTox 96 kit. Results are expressed in percentage of cytotoxicity versus time of treatment in hours. Results represent the mean ± SEM of three determinations.

**Figure 9 ijms-22-11303-f009:**
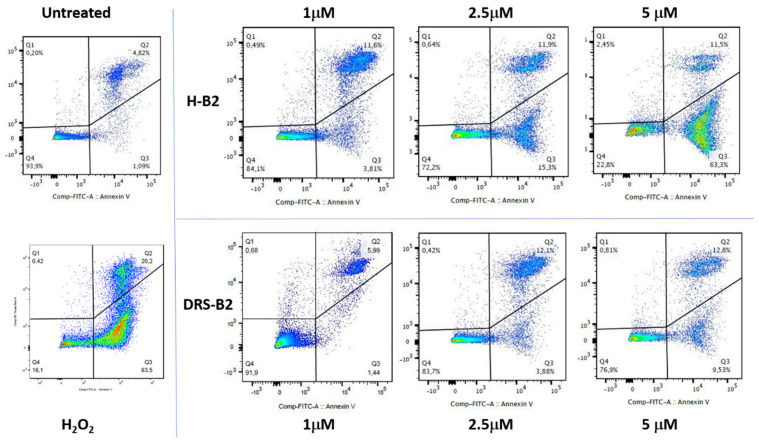
PC3 cell viability after H-B2 and DRS-B2 treatments. PC3 cells were treated with different concentrations of H-B2 or DRS-B2. H_2_O_2_ was used as a positive control for apoptosis. Twenty-four hours after treatment, cells where double stained with FITC-Annexin-V (A-V) and Propidium Iodide (PI) and analyzed by flow cytometry. The cell viability of PC3 cells was observed by measuring the amount of A-V and PI negative and positive cells. Dot plots of A-V/PI double staining after PC3 treatment with the different concentrations or H-B2 or DRS-B2 for 24 h. All experiments were performed three times.

**Figure 10 ijms-22-11303-f010:**
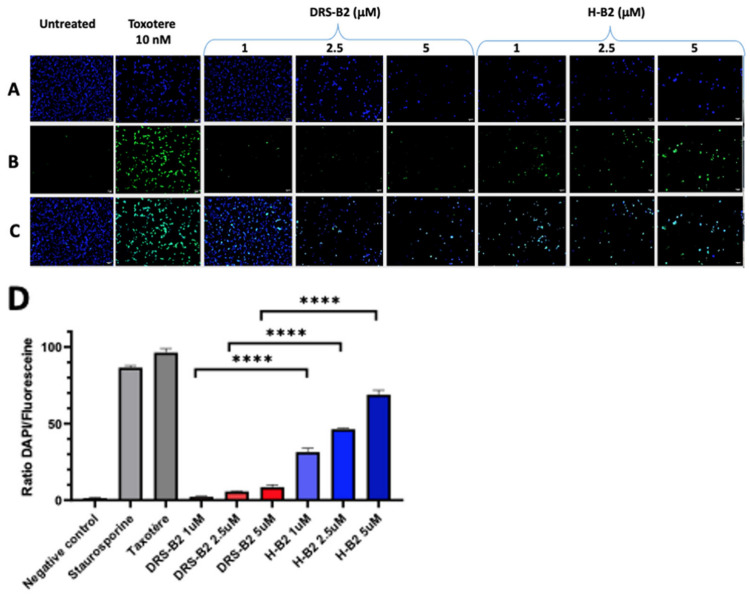
Terminal transferase-mediated dUTP nick end-labeling (TUNEL) in vitro assay of treated PC3 cells. Apoptosis of PC3 cells using TUNEL assay in vitro after DRS-B2 and H-B2 treatment. Cells were treated for 24 h with increasing concentrations of each peptide. Taxotere treatment was performed as a positive control for apoptosis. (**A**) Cells were stained with DAPI. (**B**) 3′ Hydroxyl-nucleotides of double strand DNA breaks were stained with Fluoresceine. (**C**) Merged fluorescence of DAPI and fluoresceine showing DNA fragmented cells. (**D**) Histogram representing percentage of fragmented DNA-containing cells of the total number of cells. *n* = 3 (Mean ± SD; **** *p* < 0.0001).

**Table 1 ijms-22-11303-t001:** List of cell lines and their characteristics.

Cell Line	Characteristics
PC3	Adenocarcinoma derived from bone metastasis
DU145	Adenocarcinoma derived from brain metastasis
22Rv1	Adenocarcinoma derived from a PCa primary tumor that was serially transplanted in nude mice.
BPH-1	Benign Prostatic Hyperplasia Cell Line

**Table 2 ijms-22-11303-t002:** Comparison of the IC_50_ of H-B2 and DRS-B2 extracted from Figure 1, on the different cell lines.

Cell Lines	IC_50_ (µM)	
	DRS-B2	H-B2
PC3	3.93	2.95
DU145	2.64	2.01
22RV1	2.75	2.23
BPH1	4.05	4.64

**Table 3 ijms-22-11303-t003:** Percentage of PC3 cells in each population extracted from Figure 9.

PC3 (%)	NT	H_2_O_2_	DRS-B2 (µM)			H-B2 (µM)		
			1	2.5	5	1	2.5	5
AV-/PI-	93.9	16.1	91.9	83.7	76.9	84.1	72.2	22.8
AV+/PI-	1.09	63.5	1.44	3.88	9.53	3.81	15.3	63.3
AV+/PI+	4.82	20.2	5.99	12.1	12.8	11.6	11.9	11.5
AV-/PI+	0.2	0.42	0.68	0.42	0.81	0.49	0.64	2.45

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
