# Peer review of "Antitumor Activity and Mechanism of Action of Hormonotoxin, an LHRH Analog Conjugated to Dermaseptin-B2, a Multifunctional Antimicrobial Peptide"

_ijms, 2021, doi:10.3390/ijms222111303_

Round 1

Reviewer 1 Report

Couty and colleagues provided interesting data on the chimeric compound that they designed conjugating an LHRH analogue and Dermaseptin B2. The resulting chimera H-B2 is effective in vitro and in vivo toward cancer cells and evidences have been provided demonstrating that the molecule induces apoptosis.

The paper is well written, the experimental path is clear, however few points need to be addressed in order to strengthen the results and the manuscript before publication.

-Figure 1:  the quality is not acceptable for publication and must be reprepared. Panel B and C are even illegible. 

-Figure 2: poor quality. Authors may consider to reprepare it using higher definition.

-CD studies: authors performed the assays in PBS, however, expecially at low wavelength, the chloride contained in PBS may hide partially the signal due to its high absorbance, even if proper blank has been performed. In the end, the data that author provided are convincing, but if they used PBS for a specific reason (instead for example of 10 mM sodium-phospate) this should be stated in the text.

Figure 5: poor quality. Authors may consider to re-prepare it using higher definition.

-Figure 8:  the quality is not acceptable for publication and must be reprepared. Panel A is illegible. 

-Flow cytometry analysis: in order to properly compare DRS-B2 and HB2 (and their mode of action), author should provide data using the same concentrations of both the compounds.

In all the other experiments the authors maintained a nice paralelism between the native peptide and the chimera, that however this is lacking for flowcytometry (also 5 microM DRS-B2has to be tested).

However, a parallelism would have been important to compare better not only the two molecules, but also the outputs of different experiments that converge to delineate the mode of action of the molecules.

Moreover, in the paper there is no evidence of repetition of flow-cytometry assay. Since authors used a multistep protocol, prone to variability, a single experiment is not sufficient to provide robust data. 

The repetitions of the experiment will be the perfect opportunity for authors to test also 5 microM DRS-B2 in parallel with the other already tested conditions.

-discussion, lines 340-345: a reference should be added.

(Optional): authors may consider to test the compounds (or at least the chimera) also on primary cells, in order to evaluate their capacity to discriminate healty and cancer cells. This would add value to the work but may be quite demanding, therefore this decision has to be made by authors. This could be also the subject of future investigation in another project.

Author Response

Reviewer 1:

Comments: 
-Figure 1:  the quality is not acceptable for publication and must be reprepared. Panel B and C are even illegible. 
Answer: According to both referees, we have placed panel B and C of original Figure 1 as Supplementary data Fig.S1A and S1B.
-Figure 2: poor quality. Authors may consider to reprepare it using higher definition.
Answer: According to both referees, we have modified and increased the quality of the original figure 2 and separated the doses response curves from the IC50 table.  We let the dose response as figure 2 and IC50 values presented in new Table 2.
-CD studies: authors performed the assays in PBS, however, expecially at low wavelength, the chloride contained in PBS may hide partially the signal due to its high absorbance, even if proper blank has been performed. In the end, the data that author provided are convincing, but if they used PBS for a specific reason (instead for example of 10 mM sodium-phospate) this should be stated in the text.
Answer: Usually, we used PBS as buffering solution of CD studies and we systematically subtracted the signal of PBS from the absorbance of compounds.
Figure 5: poor quality. Authors may consider to re-prepare it using higher definition.
Answer: Accordingly, we have improved the quality of the Figure 5 and assigned a new number (new Fig.6) because, as suggested by referee 2 we separated figure 4 into 2 new (figure 4 and new figure 5). 
-Figure 8:  the quality is not acceptable for publication and must be reprepared. Panel A is illegible. 
Answer: According to both referees, we have improved the quality of the original figure 8 and assigned a new number as explained above. Thus, the dot plots curves are reported in new figure 9 and the table becomes Table 3. We have also placed the histogram in the supplementary data (Fig. S2).
-Flow cytometry analysis: in order to properly compare DRS-B2 and HB2 (and their mode of action), author should provide data using the same concentrations of both the compounds.
In all the other experiments the authors maintained a nice paralelism between the native peptide and the chimera, that however this is lacking for flowcytometry (also 5 microM DRS-B2has to be tested).
Answer: In the new Figure 8, we added 1 µM and 5 µM of DRS-B2 in dot plots analysis of the manuscript and in the Table 3.
However, a parallelism would have been important to compare better not only the two molecules, but also the outputs of different experiments that converge to delineate the mode of action of the molecules.
Moreover, in the paper there is no evidence of repetition of flow-cytometry assay. Since authors used a multistep protocol, prone to variability, a single experiment is not sufficient to provide robust data. 
Answer: The flow cytometry analysis was performed three times. We have added this in the manuscript in the Figure legend.
The repetitions of the experiment will be the perfect opportunity for authors to test also 5 microM DRS-B2 in parallel with the other already tested conditions.
-discussion, lines 340-345: a reference should be added.
Answer: References (29-34) of articles describing the mechanism of action of antimicrobial peptides in general and of the dermaseptin peptide family involving a carpet model and transient pore forming have been added in the discussion part in place and location requested by the referee. 
(Optional): authors may consider to test the compounds (or at least the chimera) also on primary cells, in order to evaluate their capacity to discriminate healty and cancer cells. This would add value to the work but may be quite demanding, therefore this decision has to be made by authors. This could be also the subject of future investigation in another project.
Answer: We agree with the referee, the chimera H-B2 should be tested on primary cells to consider its effect on healthy cells and will be the subject of future investigations. However, we have previously demonstrated that DRS-B2 was inefficient on primary human prostate and skin fibroblasts (reference 11). 

Reviewer 2 Report

The submitted manuscript “Antitumor activity and mechanism of action of the Hormonotoxin, an LHRH analog conjugated to Dermaseptin-B2, a multi-functional antimicrobial peptide” by Couty et al described the development of a new targeting therapy approach by conjugating a LHRH analog with an antimicrobial peptide holding antitumor activity. This novel strategy was carefully planned and evaluated, and the results were thoroughly discussed and supported by the data presented, and substantially contribute to cancer research. I recommend the publication of this manuscript after some minor revisions that should be addressed by the authors:

  • In lines 77 and 367, please replace “anti-tumor” with “antimumor” and in line 87, “iv-vivo” with “in vivo”.

  • In lines 91 and 97, “over-expressed” should be replaced with “overexpressed”

  • In the “Materials and Methods” section, the authors report that “the peptide is dissolved in sterile water for experimental use”. However, secondary structure of HB-2 and DRS-B2 was assessed in PBS and DPS. Was the secondary structure assessed in water as well? Also, is the chimeric peptide stable in water? For how long? Was this checked? Is the antimicrobial peptide fully active dissolved in water? It would be important to consider the comparation the stability of the chimeric peptide and the activity of the peptide in water and in PBS.

  • Regarding Figure 1, the RP-HPLC profile and the ESI-MS spectra of the synthetic H-B2 are too small to be perceptible. For this reason, I would suggest that both of them would be reported as supplementary data. Also, the resolution of the primary structure of the hormonotoxin should be improved.

  • Figure 2 should be divided so that the graphics are presented as the new Figure 2 and the table as Table 1. Also, please remove the “sans titre” box of the image, increase the size of the graphics and improve their resolution.

  • For a better reading, I suggest that Figure 4 included only the graphics regarding the “effect of tumor volume versus time of treatment” and the “tumor weight after sacrifice of mice”. PC3 tumor proliferation evaluation should be presented in a separated figure.

  • Please rearrange Figure 8, so the dot plots of the controls and the increasing concentrations of H-B2 are all the same size and more perceptive. Also, as the information reported in the histogram is the same as the information in the table, the histogram could be presented as supplementary figure and the table could be presented as a new Table, and not as a figure.

  • The contrast in figures 9A, B and C should be improved.

  • Finally, as the LHRH-receptor is overexpressed in other cancer cells, such as endometrial and ovarian cell lines, have you tested this new hormonotoxin in tumor cell lines other than prostate cell lines? If so, this should be mentioned in the manuscript.

Author Response

Reviewer 2:

•    In lines 77 and 367, please replace “anti-tumor” with “antimumor” and in line 87, “iv-vivo” with “in vivo”.
Answer: done
•    In lines 91 and 97, “over-expressed” should be replaced with “overexpressed”
Answer: done
•    In the “Materials and Methods” section, the authors report that “the peptide is dissolved in sterile water for experimental use”. However, secondary structure of HB-2 and DRS-B2 was assessed in PBS and DPS. Was the secondary structure assessed in water as well? Also, is the chimeric peptide stable in water? For how long? Was this checked? Is the antimicrobial peptide fully active dissolved in water? It would be important to consider the comparation the stability of the chimeric peptide and the activity of the peptide in water and in PBS.
Answer: Peptides were dissolved in water for in vitro biological studies and in adequate buffer for in vivo and structural analysis. This was specified in the Materials and Methods section.
•    Regarding Figure 1, the RP-HPLC profile and the ESI-MS spectra of the synthetic H-B2 are too small to be perceptible. For this reason, I would suggest that both of them would be reported as supplementary data. Also, the resolution of the primary structure of the hormonotoxin should be improved.
Answer: See response to referee 1, we have corrected accordingly. The primary structure of the synthetic Hormonotoxin is reported in a new figure 1 and the RP-HPLC profil and ESI-MS spectra are placed in the supplementary data as figure S1A and S1B respectively.
•    Figure 2 should be divided so that the graphics are presented as the new Figure 2 and the table as Table 1. Also, please remove the “sans titre” box of the image, increase the size of the graphics and improve their resolution.
Answer: See response to referee 1, we have corrected accordingly. Dose response curves were presented in new figure 2 with high resolution and IC50 were reported in a Table 2.
•    For a better reading, I suggest that Figure 4 included only the graphics regarding the “effect of tumor volume versus time of treatment” and the “tumor weight after sacrifice of mice”. PC3 tumor proliferation evaluation should be presented in a separated figure. 
Answer: Figure 4 was modified accordingly with only tumor volume vs time of treatment and tumor weight after sacrifices. PC3 tumor proliferation and Ki67 staining were reported on a new figure 5.
•    Please rearrange Figure 8, so the dot plots of the controls and the increasing concentrations of H-B2 are all the same size and more perceptive. Also, as the information reported in the histogram is the same as the information in the table, the histogram could be presented as supplementary figure and the table could be presented as a new Table, and not as a figure.
Answer: We have modified accordingly, see response to referee 1.
•    The contrast in figures 9A, B and C should be improved.
Answer: We have improved the contrast of each figure 9A, B and C and assigned a new number as explained above (new Fig. 10A, B, and C). We have also added a 5 µM concentration of DRS-B2 (Figure 10).
•    Finally, as the LHRH-receptor is overexpressed in other cancer cells, such as endometrial and ovarian cell lines, have you tested this new hormonotoxin in tumor cell lines other than prostate cell lines? If so, this should be mentioned in the manuscript. 
Answer: Thank you for your comment which is relevant, we have not yet tested the hormonotoxin on the two types of cancer overexpressing the LHRH receptor but we will certainly integrate it in our next study.